# Putative Clinical Potential of *ERBB2* Amplification Assessment by ddPCR in FFPE-DNA and cfDNA of Gastroesophageal Adenocarcinoma Patients

**DOI:** 10.3390/cancers14092180

**Published:** 2022-04-27

**Authors:** Elisa Boldrin, Marcodomenico Mazza, Maria Assunta Piano, Rita Alfieri, Isabella Monia Montagner, Giovanna Magni, Maria Chiara Scaini, Loretta Vassallo, Antonio Rosato, Pierluigi Pilati, Antonio Scapinello, Matteo Curtarello

**Affiliations:** 1Immunology and Molecular Oncology, Veneto Institute of Oncology IOV-IRCCS, Via Gattamelata 64, 35128 Padova, Italy; elisa.boldrin@iov.veneto.it (E.B.); mariaassunta.piano@iov.veneto.it (M.A.P.); mariachiara.scaini@iov.veneto.it (M.C.S.); antonio.rosato@iov.veneto.it (A.R.); 2Surgical Oncology of the Esophagus and Digestive Tract Unit, Veneto Institute of Oncology IOV-IRCCS, Via dei Carpani 16, 31033 Castelfranco Veneto, Italy; marcodomenico.mazza@iov.veneto.it (M.M.); rita.alfieri@iov.veneto.it (R.A.); pierluigi.pilati@iov.veneto.it (P.P.); 3Pathology Unit, Veneto Institute of Oncology IOV-IRCCS, Via dei Carpani 16, 31033 Castelfranco Veneto, Italy; isabellamonia.montagner@iov.veneto.it (I.M.M.); loretta.vassallo@aulss8.veneto.it (L.V.); antonio.scapinello@iov.veneto.it (A.S.); 4Clinical Research Unit, Veneto Institute of Oncology IOV-IRCCS, Via Gattamelata 64, 35128 Padova, Italy; giovanna.magni@iov.veneto.it; 5Oncology and Immunology Section, Department of Surgery Oncology and Gastroenterology, University of Padova, Via Gattamelata 64, 35128 Padova, Italy

**Keywords:** liquid biopsy, cell-free DNA (cfDNA), HER2, *ERBB2*, gastroesophageal adenocarcinoma, droplet digital PCR (ddPCR)

## Abstract

**Simple Summary:**

Gastroesophageal adenocarcinoma (GEA) has a poor prognosis. However, since the HER2 positive subgroup could benefit from trastuzumab targeted therapy, considerable effort has been spent in determining the HER2 status in these patients. To date, immunohistochemistry and in situ hybridization are the gold standard methods for assessing HER2/*ERBB2* overexpression/amplification in tumor specimens. However, they have several limitations due to their cost, the large number of undetermined cases, and the impossibility of longitudinal patient monitoring. Here, we report the potential of a molecular method (droplet digital PCR) to investigate *ERBB2* status in both solid and liquid biopsies of GEA. Results suggest that this methodology could be used to implement current histological analysis in solid biopsy and that it may be feasible in liquid biopsy. An alternative, more sensitive method of assessing HER2 status may aid physicians in their therapeutic decision-making, benefiting the patient. Liquid biopsy could also overcome the limitations of tissue-based analyses.

**Abstract:**

Anti-HER2 monoclonal antibody trastuzumab improves the survival of those patients with advanced gastroesophageal adenocarcinoma (GEA) exhibiting HER2/*ERBB2* overexpression/amplification. The current gold standard methods used to diagnose the HER2 status in GEA are immunohistochemistry (IHC) and silver or fluorescence in situ hybridization (SISH or FISH). However, they do not permit spatial and temporal tumor monitoring, nor do they overcome intra-cancer heterogeneity. Droplet digital PCR (ddPCR) was used to implement the assessment of HER2 status in formalin-fixed paraffin-embedded (FFPE) tumor DNA from a retrospective cohort (86 patients) and in cell-free DNA (cfDNA) samples from a prospective cohort (28 patients). In comparison to IHC/SISH, ddPCR assay revealed *ERBB2* amplification in a larger patient fraction, including HER2 2+ and 0–1+ of the retrospective cohort (45.3% vs. 15.1%). In addition, a considerable number of HER2 2+ and 0–1+ prospective patients who were negative in FFPE by both IHC/SISH and ddPCR, showed *ERBB2* amplification in the cfDNA collected just before surgery. cfDNA analysis in a few longitudinal cases revealed an increasing *ERBB2* trend at progression. In conclusion, ddPCR in liquid biopsy may improve the detection rate of HER2 positive patients, preventing those patients who could benefit from targeted therapy from being incorrectly excluded.

## 1. Introduction

Gastric adenocarcinoma (GAC), esophagogastric junction adenocarcinoma (EGJA), and esophageal adenocarcinoma (EADC) are collectively termed gastroesophageal adenocarcinomas (GEAs). Globally, GAC ranks fifth for incidence and fourth for mortality. The incidence and mortality rates of noncardia GAC are declining, while these parameters are increasing for the gastric cardia junction [1]. Moreover, esophageal cancer is the seventh tumor in overall incidence, and EADC is increasing at a faster rate than the squamous subtype in high-income countries as a result of increased body mass index and gastroesophageal reflux [2].

Although surgery remains the cornerstone for the treatment of locally advanced GEAs, the introduction of neoadjuvant treatments has significantly improved overall survival (OS) and disease-free survival (DSF) [3,4].

The Cancer Genome Atlas (TCGA) project has recently classified GAC into four different molecular subtypes: Epstein-Barr virus positive (EBV, 9%), microsatellite unstable (MSI, 22%), chromosomal unstable (CIN, 50%), and genomic stable (GS, 20%) [5]. Each tumor subtype can be localized throughout the anatomic sites of the stomach (cardia, fundus, body, and antrum). Moreover, TCGA has shown that EADC and EGJA are both molecularly similar to the GAC CIN phenotype [6]. 

Following the TCGA classification, Sohn et al. investigated the relationship between tumor subtypes and clinical outcome. They found that the EBV subtype had the best prognosis, while GS had the worst [7].

Genomic studies enabled a better understanding of GAC molecular pathogenesis and, above all, indicated the possibility of using targeted therapies in GEAs. Among the several genetic alterations that characterize GEAs, the overexpression/amplification of HER2/*ERBB2* has been found to be one of the most frequent (7–38%), especially in the CIN subtype [5,8,9,10]. 

Based on the TCGA classification, Gonzalez et al. showed that immunohistochemistry (IHC) can categorize GAC subtypes and evaluate HER2 expression in formalin-fixed paraffin-embedded (FFPE) specimens [11]. To date, the National Comprehensive Cancer Network Clinical Practice Guidelines in Oncology (NCCN Guidelines) recommend IHC as the gold standard for determining HER2 status in GEAs [12]. According to these guidelines, tumor specimens are classified as HER2 3+ (positive), 2+ (equivocal), or 0–1+ (negative), based on the staining intensity. No further testing is required if the staining is negative or positive, while in equivocal cases (i.e., HER2 2+), subsequent testing using an in situ hybridization (ISH) method should be performed to confirm the *ERBB2* status (i.e., the gene that encodes HER2) [12]. 

So far, data discrepancy exists about HER2 targeted therapy efficacy. Indeed, while the ToGA trial showed that the monoclonal antibody trastuzumab significantly prolongs OS in inoperable locally advanced, and metastatic HER2 positive GEA [8], other randomized phase III trials using HER2-targeted therapies other than trastuzumab did not find any benefits for OS [13,14,15,16]. New HER2-targeted agents were developed in view of this discrepancy. One of the most promising treatments is the antibody drug-conjugated trastuzumab deruxtecan (T-DXd) that showed antitumor activity also in patients with low levels of HER2 expression and prolonged OS in advanced HER2 positive gastric cancer resistant to other therapies, including trastuzumab [17,18].

The discordance in the efficacy of HER2 targeted therapy in GEAs may reflect the increased heterogeneity of HER2 positivity compared to breast cancer [19,20]. A high variability across tissue sections and a frequent discordance between endoscopic biopsies and surgical specimens have also been reported [10]. Furthermore, a recent prospective multicenter study (VARIANZ; NCT02305043), which aimed to better define the efficacy of trastuzumab, pointed out the relevance of setting up an appropriate threshold of HER2/*ERBB2* expression/amplification in tissue testing. The study highlighted that the response rate to targeted therapy is strongly conditioned by HER2 expression heterogeneity in GEAs [21]. Thus, it seems that HER2 status assessment with a traditional tissue-based method may provide insufficient information to determine the eligibility of patients for trastuzumab due to temporal and spatial intra-tumor heterogeneity [22].

Liquid biopsy and, in particular, plasma cell-free DNA (cfDNA) analysis was demonstrated to give a more comprehensive characterization of the tumor molecular alterations compared to a single or few solid biopsies and proved to be a valid tool to overcome intra-tumor heterogeneity [23,24,25]. Moreover, the reduced invasiveness of liquid biopsy allows for the monitoring of tumor progression and the efficacy of new putative druggable targets through serial blood sampling [25,26,27,28,29]. The potential clinical application of cfDNA is rapidly growing in pulmonary, colorectal, and breast cancer [30,31,32], but its use has not yet entered into the routine of GEA patient management. 

To date, only a few studies have investigated the feasibility of detecting *ERBB2* amplification in the liquid biopsy of GEA patients [33,34,35]. The authors used droplet digital PCR (ddPCR) as the molecular method to detect *ERBB2* status in liquid biopsy. High *ERBB2* copy number variation (CNV) in cfDNA was associated with a shorter survival [33], and an increase in *ERBB2* CNV was detected at the time of tumor progression [34]. These studies highlighted the putative relevance of a correct assessment of *ERBB2* amplification for patient stratification and therapy decision-making.

Here, FFPE specimens of a retrospective cohort of GEA patients were evaluated using ddPCR to detect *ERBB2* amplification with the aim of better defining the HER2 status. The analysis was also extended to cfDNA samples of a GEA prospective cohort in order to evaluate the impact of cfDNA analysis on *ERBB2* amplification detection.

## 2. Materials and Methods

### 2.1. Patients

Eighty-six retrospective and twenty-eight prospective GEA patients were enrolled for this study among those patients who were referred to the Surgical Oncology of the Esophagus and Digestive Tract Unit of the Veneto Institute of Oncology (IOV-IRCCS, Padova, Italy). For both cohorts, the patients’ inclusion criteria were: (i) ≥18 years of age; (ii) histological diagnosis of GEA (all stages); and (iii) availability of FFPE tumor specimen at diagnosis and/or at surgery resection. Exclusion criteria were the concurrent diagnosis of synchronous or metachronous tumors within five years prior to the diagnosis of GEA. 

Retrospective FFPE samples were retrieved from the archives of the IOV-IRCCS Pathology Unit (2016–2019). Patients from the prospective cohort were enrolled between 2019–2020 and for all of them a blood sample was collected just before surgery. Some of the patients were also followed longitudinally.

This study was approved by the IOV-IRCCS Comitato Etico per la Sperimentazione Clinica (CESC) (cod. number CESC IOV: 2019/72) and was carried out in accordance with the Code of Ethics of the World Medical Association (Declaration of Helsinki and its later amendments). All the people who participated in the study gave their written informed consent in accordance with the Helsinki Declaration.

### 2.2. Immunohistochemistry

All patients were tested for DNA mismatch repair (MMR) proteins (MLH1, MSH2, MSH6, and PMS2) normal expression, p53 alterations, and the presence of the Epstein–Barr virus early RNA (EBER) antigen.

Additionally, HER2 protein expression in GEA specimens was assessed by IHC using the rabbit monoclonal antibody Ventana anti-HER2/neu (4B5) (Roche, Monza, Italy). Each specimen was stained with hematoxylin and eosin (H&E). Staining was carried out in 4 μm FFPE tissue sections. HER2 status was considered positive with HER2 3+, negative with HER2 0 or 1+, and equivocal with HER2 2+.

In GEA specimens with equivocal HER2 status, a dual-color silver in situ hybridization (SISH) was performed using the Ventana INFORM HER2 Dual ISH DNA probe Cocktail assay (Roche, Monza, Italy). SISH was carried out using a dinitrophenyl labeled probe specific for *ERBB2* and a digoxigenin labeled probe specific for chromosome 17 (Chr 17). Each staining had an internal positive and negative control on the same slide. Signal visualization in cell nuclei was assessed by light microscopy, where *ERBB2* appeared as a black signal and Chr 17 centromer as a red signal. Using a 60× magnification, a manual count of the number of *ERBB2* and Chr 17 signals was conducted in 20 non-overlapping invasive cancer cell nuclei, and the *ERBB2*/Chr 17 ratio was then recorded. *ERBB2* was considered amplified if the *ERBB2*/Chr 17 ratio was ≥2.0, or not amplified if the ratio was <2. An additional 20 cells were investigated if a case had an average *ERBB2* signal per cell < 4.0 or ≥6 with a *ERBB2*/Chr 17 ratio ≥ 2.0 or <2.0, respectively [36].

The pattern was evaluated by a senior pathologist in accordance with the ASCO/CAP protocol for IHC interpretation.

### 2.3. DNA Extraction

FFPE-DNA was isolated from tumor specimen using the QIAamp Mini Kit (Qiagen, Milan, Italy) from eight consecutive 10 µm thick sections, according to manufacturer instructions. A neoplastic component ≥ 70% was considered adequate for tumor-DNA analysis, while manual macro-dissection tumor enrichment was performed for specimens with a lower neoplastic amount. DNA quantity and quality were assessed using the NanoDrop 1000 spectrophotometer (Thermo Fisher Scientific, Milan, Italy). 

Peripheral blood samples were collected in cell-free DNA BCT tubes (Streck, La Vista, NE, USA). The plasma was centrifuged at 2000× *g* and subsequently at 16,000× *g* to separate it from corpuscular components and to remove cellular debris, and then stored at −80 °C. The Maxwell RSC ccfDNA Plasma Kit (Promega, Milan, Italy) was used to extract cfDNA from 1 mL of plasma. 

The quantity of cfDNA was assessed with the Qubit dsDNA HS Assay kit (Thermo Fisher Scientific, Milan, Italy); cfDNA quantity ranged between 6 and 30 ng/mL of plasma. Furthermore, the Agilent Tape Station 2200 was used to assess the quality of cfDNA samples. This was carried out by means of the cfDNA screen tape assay kit (Agilent Technologies, Milan, Italy). A representative image of the quality of some cfDNA samples showed that the majority of them was suitable for subsequent analyses (Figure A1).

### 2.4. ddPCR

The amplification of the *ERBB2* target gene in FFPE-DNA and cfDNA was analyzed by ddPCR using *RPPH1* as reference gene. The ddPCR target and reference gene assays were run in duplicate. 

ddPCR amplification analysis was carried out in a 20 μL reaction mixture containing 10 μL of 2× ddPCR Supermix for probes (no dUTP) (Bio-Rad, Milan, Italy), 1 µL of 20× target *ERBB2* primers/probe (FAM) (ddPCR™ Copy Number Variation Assays Validated, Assay ID: dHsaCP1000116; Bio-Rad, Milan, Italy), and 1 µL of 20× reference custom *RPPH1* primers/probe (HEX) (ddPCR™ Copy Number Variation Assays custom; Bio-Rad, Milan, Italy). Primers and probe sequences of *RPPH1* were previously described by Shoda K. et al. [22]. Each pair of primers/probes was in a final concentration of 900 nM/250 nM in the reaction mixture. Equal aliquots of the master mix reaction were dispensed into the reaction tubes, and the sample template was added. PCR amplification was performed in two replicates, with 15 ng/well for FFPE-DNA and 0.7 ng/well for cfDNA. The 20 µL mixture of reaction mix and DNA were loaded into a DG8™ droplet generator cartridge with 70 µL of droplet generator oil for probes (Bio-Rad, Milan, Italy). Droplets were generated by a QX200 droplet generator device (Bio-Rad, Milan, Italy) and carefully transferred to a 96-well PCR plate to perform PCR. PCR conditions were: enzyme activation at 95 °C for 10′, followed by 40 cycles at 94 °C for 30″ (denaturation), 60 °C for 1′ (annealing and extension), and 98 °C for 10′ (enzyme inactivation). After PCR, the droplets were read with the QX200 droplet reader and analyzed with the QuantaSoft™ version 1.7.4 software (Bio-Rad, Milan, Italy).

### 2.5. ERBB2 CNV Evaluation by ddPCR

The software determined the copy number using the formula: CNV = A/B × N, where A and B are the concentrations of the target molecule and the reference molecule, respectively; N is the number of copies of the reference loci in the genome (usually 2). 

To set up the positivity cut-off value for *ERBB2* amplification in FFPE-DNA and cfDNA, 10 normal mucosa FFPE specimens and 10 healthy volunteer plasma samples (controls) were analyzed, respectively. The cut-off value was calculated as the CNV mean of controls + 2SD. Based on this calculation, we considered FFPE-DNA samples with a CNV > 3.5 and cfDNA samples with a CNV > 3.7 as positive for *ERBB2* amplification.

Reactions that generated a low number of droplets (<10,000 per 20 μL PCR) were excluded from further analysis. The individual droplet (individual partition) volume was 1 nL. When applicable, ddPCR experiments were designed, performed, and reported in line with the Digital MIQE Guidelines [37].

### 2.6. Statistics

Statistical analyses were performed using SigmaPlot version 14.0 (Systat Software Inc., San Jose, CA, USA). A Wilcoxon Mann-Whitney test or a Student’s *t*-test was performed to compare CNV values between groups. All of the tests were two-tailed. The difference between CNV values of HER2 positive vs the CNV values of HER2 negative patients was determined by a receiver operating characteristic (ROC) curve using the GraphPad Prism software (version 6.0 for Windows, San Diego, CA, USA). A *p*-value < 0.05 was considered statistically significant.

## 3. Results

### 3.1. HER2/ERBB2 Status Analysis in Solid Biopsy

#### 3.1.1. Clinicopathologic Characteristics of the Retrospective Cohort

The FFPE samples of 86 GEA patients were retrospectively retrieved in order to assess ddPCR performance for *ERBB2* amplification.

The clinicopathologic characteristics of patients are described in Table 1. Briefly, the median age at diagnosis was 76 years (range: 44–97), and males were more represented than females (60% vs. 40%, respectively). In addition, I/II stage tumors were more frequent than III/IV stage tumors (56% vs. 44%); GEA subgroups, determined by IHC typing, were GS (47%), CIN (33%), MSI (17%), and EBV+ (3%).

With regard to the HER2 status, considering the IHC and SISH assays together, the total number of HER2 positive and HER2 negative patients was 13/86 (15.1%) and 73/86 (84.9%), respectively (Table 1). A representative IHC and SISH staining was reported in Figure A2.

#### 3.1.2. HER2/*ERBB2* Status Analysis by ddPCR in the Retrospective Cohort

In order to evaluate *ERBB2* amplification (i.e., copy number variation, CNV), ddPCR analysis was performed in DNA extracted from the same tissue block previously analyzed by IHC. A CNV value of >3.5, which was calculated by analyzing a series of normal FFPE mucosa (see Section 2), was used as cut-off for positivity.

As expected, all of the HER2 3+ samples resulted positive for *ERBB2* amplification (11/11 = 100%). When we examined the HER2 2+ samples, we found that, next to the 2 positive samples detected by SISH, 2 additional cases had *ERBB2* amplified (4/7 = 57.1%); more surprisingly, among the 68 HER2 0–1+, 24 were amplified (24/68 = 35.2%). Thus, 39/86 patients (45.3%) resulted positive for *ERBB2* amplification vs. 13/86 (15.1%) found by IHC/SISH (Table 2).

While these results show total agreement between IHC and ddPCR in samples highly positive for *ERBB2* amplification (i.e., HER2 3+), they highlight the greater sensitivity of ddPCR vs. IHC/SISH in samples with a low amplification level. Indeed, ddPCR detects a considerable number of *ERBB2* amplified cases, in the HER2 0–1+ subgroup, while, as expected, there is a proportional decrease in the relative frequencies of *ERBB2* amplified cases along with the reduction in the IHC score intensity (3+, 2+, and 0–1+). The data suggest that IHC alone underestimates the number of HER2 positive cases.

When only ddPCR *ERBB2* amplified cases in the different subgroups were considered, the median CNV value found in HER2 3+ (10.9, range 3.8–121) was statistically different from HER2 2+ (4.29, range 3.6–5.0; *p* = 0.031) and HER2 0–1+ (4.53, range 3.55–7.4; *p* < 0.001). Interestingly, HER2 2+ and HER2 0–1+ groups showed similar CNV positive median values (*p* = 0.599) (Figure 1).

Due to the high number of *ERBB2* amplified cases identified with the use of ddPCR in the 24 HER2 0–1+ patients, we decided to analyze them using SISH since they had previously been excluded in accordance with current diagnostic guidelines. Of these, 7 were excluded due to the inadequacy of the FFPE sample. Among the remaining 17 samples, 5 were *ERBB2* amplified (29.4%; Table A1). Thus, SISH data concerning HER2 0–1+ samples indicate that they can be misclassified by IHC alone. 

The similarity in *ERBB2* amplification by ddPCR between HER2 2+ and HER2 0–1+ subgroups suggests that HER2 0–1+ should also be considered “equivocal” like HER2 2+, and additional investigations, such as ddPCR, should be performed to resolve this ambiguity.

The accuracy of ddPCR analysis in the identification of samples positive for *ERBB2* amplification was evaluated by drawing a receiver operating characteristics (ROC) curve. HER2 positive or negative status was assigned based on the combination of IHC/SISH results (“standard method”) and ROC curve was performed by comparing the CNVs of the HER2 positive to the ones of the HER2 negative samples. The area under the curve (AUC) was 0.909 ± 0.041 (Std Error = 0.04091; 95% CI 0.8288–0.9892, *p* < 0.0001; Figure 2). The sensitivity and the specificity of the test resulted 94.44% and 62.07%., respectively, by assigning a cut-off value of 3.65, very close to the cut-off calculated by analyzing the normal FFPE mucosa samples.

### 3.2. ERBB2 Status Analysis in Liquid Biopsy

#### 3.2.1. Clinicopathologic Characteristics of the Prospective Cohort

Once we characterized the retrospective cohort for *ERBB2* amplification using ddPCR, we applied this analysis to the 28 patients belonging to the prospective cohort. The clinicopathologic characteristics of patients were comparable to those of the retrospective cohort. Briefly, the median age of patients was 67 years (range 40–96) and males, as in the retrospective cohort, were more represented (60.7%). As regards molecular subtype distribution, GS was the most representative (64.3%), followed by CIN (28.6%), and MSI (7.1%). Tumors were mainly stage III/IV (67.9%; Table 3). 

HER2 status, evaluated at surgery by IHC, classified prospective patients as: HER2 3+ (5/28 = 17.9%), HER2 2+ (7/28 = 25%), and HER2 0–1+ (16/28 = 57.1%). SISH was also performed for the seven HER2 2+ equivocal cases, but none of them were *ERBB2* amplified. Thus, the percentage of HER2 negative cases was 82.1% (Table 3).

#### 3.2.2. Performance Analysis of ddPCR by Mimicking Liquid Biopsy Conditions

Before starting the ddPCR analysis in the prospective cohort, we evaluated the performance of the assay in conditions that mimic those present in the liquid biopsy. As tumor-DNA fraction is usually present at a very low concentration in the cfDNA extracted from plasma, conditions similar to those of circulating tumor DNA were simulated.

First, to mimic the very low quantity of tumor DNA in plasma, the tumor FFPE-DNA of a HER2 3+ patient (*ERBB2* CNV = 100) was serially diluted with water and then analyzed by ddPCR. For PCR amplification, the total undiluted tumor DNA amount was 32 ng and the dilutions ranged from 1:2 to 1:256. The CNV was detectable and constant up to the 1:64 dilution, corresponding to 0.5 ng of tumor DNA. In subsequent dilutions, CNV was still detectable, but with high uncertainty (data not shown).

Since a substantial fraction of cfDNA may be represented by normal DNA that can interfere with the detection of tumor DNA during CNV normalization, another assay was performed by diluting tumor DNA with its normal counterpart. In this assay, the sample remained still above the threshold of positivity up to a dilution of 1:64, corresponding to 0.5 ng of tumor DNA vs. 31.5 ng of normal DNA (1.5% tumor fraction; Figure 3).

#### 3.2.3. Liquid Biopsy Analysis of cfDNA Point at the Time of Surgery

Following the assessment of ddPCR performance for *ERBB2* detection, FFPE-DNA isolated from the tumor surgical specimens and cfDNA samples collected immediately before surgery from the 28 prospective patients were analyzed.

By analyzing FFPE-DNA from tumor specimens with ddPCR, *ERBB2* amplification was found in 7/28 (25%) patients; 5 in HER2 3+, 1 in HER2 2+, and 1 in HER2 0–1+ (Table 4). This data showed a high concordance between IHC and ddPCR in FFPE. In contrast, when we analyzed the time-matched cfDNA, considering a CNV cut-off value of >3.7 (see Materials and Methods), *ERBB2* amplification was also found in several HER2 negative patients (HER2 2+/SISH^neg^ and HER2 0–1+): 5/7 (71.4%) in HER2 2+ and 11/16 (68.8%) in HER2 0–1+. As expected, all of the HER2 3+ samples were amplified in both ddPCR FFPE-DNA and cfDNA (Table 4). The data show that ddPCR in cfDNA detects a higher number of patients with *ERBB2* amplification than IHC/SISH. This suggests that liquid biopsy may better depict the whole tumor genetics than a single solid biopsy, overcoming the high intra-tumor heterogeneity of GEAs.

CNV values obtained from FFPE-DNA and cfDNA time-matched samples were compared to investigate the differences in *ERBB2* detection between solid and liquid biopsies (Figure 4). 

As reported in Figure 4A, cfDNAs of HER2 positive group (HER2 3+) exhibited a mean CNV value lower than the mean CNV value found in FFPE-DNAs, although still above the cut-off of positivity. This difference was not statistically significant (mean: 4.3 ± 0.59 vs. 18.1 ± 13.01; *p* = 0.077; Figure 4A). Considering the great intra-tumor heterogeneity of HER2 expression in GEA, the lower CNV mean value observed in cfDNA could be explained by the co-presence of tumor DNA derived from HER2 positive as well as HER2 negative cells. 

By contrast, when comparing *ERBB2* amplification in FFPE-DNA and cfDNA in the HER2 negative group (HER2 2+/SISH^neg^ and HER2 0–1+), the difference between CNV values was statistically significant with a *p*-value of <0.001 (median CNV value: 2.0; range: 0.9–18.4 vs. 4.5; range: 2.7–5.8, respectively; Figure 4B). The greater positivity rate in the cfDNA vs FFPE samples again suggests that liquid biopsy could be a better approach for the detection of *ERBB2* status than the single FFPE biopsy.

#### 3.2.4. Longitudinal Liquid Biopsy Analysis

To verify whether *ERBB2* amplification in cfDNA samples could be useful to monitor patients during their follow-up, we analyzed longitudinally collected cfDNAs of seven patients (G005, G013, G015, G016, G025, G030, and A564), representative of different clinical history. Samples were collected at diagnosis and/or surgery and at each follow-up visit after tumor resection. Four out of seven patients (G016, G025, G030, and A564) showed progression within the first twelve months after surgery, while three patients (G005, G013 and G015) did not relapse and they can be defined as “non-progressors”.

As concern patients who progressed within 12 months after surgery:

Patient G016, who was not eligible to undergo neoadjuvant therapy for cardiac comorbidities, had a pT3N3 HER2 3+ GEA (GS) tumor at the time of surgery. Despite the HER2 positive phenotype, this patient did not receive trastuzumab-based adjuvant therapy for the same comorbidities described above. The cfDNA collected immediately before surgery (1st cfDNA point) had *ERBB2* amplification, as did its time-matched FFPE surgical specimen. The cfDNA sample collected three months (2nd cfDNA point) after surgery was still amplified and remained positive until the time of progression (lung nodule) after twelve months (4th cfDNA point). The tumor soluble carcinoembryonic antigen (S-CEA) was positive at every time point, whereas the soluble carbohydrate antigen 19.9 (S-CA 19.9) became positive at the time of metastasis (Figure 5).

Patient G025 presented with a ypT4aN3a HER2 2+/SISH^neg^ GEA (GS) at the time of surgery, which was performed after neoadjuvant therapy. The cfDNA collected before surgery (1st cfDNA point) was *ERBB2* amplified, while the corresponding FFPE surgical specimen was negative. Subsequent cfDNA samples were all positive with an increasing trend, and the patient progressed with bone metastasis nine months after surgery. Both S-CEA and S-CA 19.9 were always negative (Figure 5).

Patient G030 had a ypT3N3b HER2 0+ GEA (CIN) at the time of surgery. The patient received neoadjuvant therapy and adjuvant treatment after surgery. cfDNA samples were always positive (*ERBB2* amplified) except at the six-month point (3rd cfDNA), which was borderline. The patient progressed fourteen months after surgery. S-CEA was always under the cut-off for positivity, while S-CA 19.9, positive at surgery, decreased during adjuvant treatment but increased at progression (Figure 5). 

Patient A564 was diagnosed with a cT3N1 HER2 2+/SISH^neg^ GEA (CIN). At surgery, after neoadjuvant therapy, the patient presented as an ypT3N3. At the time of diagnosis, the cfDNA was positive for *ERBB2* amplification, while it was borderline at surgery, however, the FFPE was positive for amplification. The cfDNA sample was still borderline three months after surgery, and the cfDNA samples were positive six months after surgery and at progression. At progression, the S-CEA was borderline and the S-CA 19.9 was negative (Figure 5). 

Among the patients defined as “non-progressors”:

Patient G005 had a ypT3N3a HER2 0+ GEA (GS) at the time of surgery. The patient received neoadjuvant therapy and adjuvant treatment after surgery. FFPE-DNA resulted not amplified for *ERBB2* in ddPCR. On the contrary, the time matched cfDNA was highly positive, as well as at three months after surgery. However, the *ERBB2* CNV decreased to a value very close to the cut-off at twelve months. The patient did not show any clinical signs of relapse (Figure 6).

Patient G013 (pT1aN0) was classified as HER2 3+ GEA (CIN) at the time of surgery. The *ERBB2* positivity was also confirmed by ddPCR in FFPE-DNA. The time-matched cfDNA showed a positive *ERBB2* CNV value very close to the cut-off, suggesting that the solid biopsy was not representative of the whole tumor genetics. CNV value was stable at three months and resulted under the cut-off at twelve months (Figure 6).

Patient G015 (pT2N0) was diagnosed as HER2 0+ GEA (GS) at the time of surgery and the FFPE-DNA was negative for *ERBB2* by ddPCR. The *ERBB2* CNV value trend in cfDNA samples was similar to that observed in patient G005 during the 12 months after surgery (Figure 6).

Overall, these data highlight that *ERBB2* amplification detected in longitudinally collected cfDNA samples might reflect the clinical evolution of the disease.

## 4. Discussion

Gastroesophageal adenocarcinomas (GEA) have a very poor prognosis and, in particular the incidence of cardia/junction tumors and EADCs is increasing [1,38]. In recent years, much effort has been made to molecularly characterize GEA patients with the aim of better stratifying them for targeted therapy.

Although data on the benefit of HER2 targeted therapy are still being debated [8,39], trastuzumab is approved for the first-line treatment of inoperable locally advanced and metastatic HER2 positive gastric cancer. This treatment partial failure is probably due to an inappropriate HER2 positivity detection and to the heterogeneity of HER2 expression.

The current histological detection method (i.e., IHC/SISH) for HER2/*ERBB2* assessment seems to be somewhat inadequate, considering in particular the literature focused on *ERBB2* amplification detection by new molecular techniques. Indeed, despite several studies observed a high concordance between HER2 expression evaluated by IHC/SISH and *ERBB2* amplification assessed by NGS in GEA [40,41,42], a recent work found that the anti-HER2 therapy outcome was worse in terms of OS and time to treatment discontinuation (TTD) in GEA patients classified as HER2 positive by IHC/FISH but negative for *ERBB2* amplification by NGS [43]. Moreover, this study showed that a higher *ERBB2* CNV was associated with a longer OS and TTD. In addition, this feature seems also to be associated with a longer progression free survival (PFS), and, thus, it could be a significant predictor of anti-HER2 treatment efficacy [44]. The inadequacy of HER2 assessment based exclusively on IHC/SISH was pointed out by recent data from the VARIANZ study showing that different laboratories do not always agree on how the HER2 status should be assessed, particularly for the intermediate HER2 scoring [21]. Recent data from the VARIANZ study showed that different laboratories do not always agree on how the HER2 status should be assessed, particularly for the intermediate HER2 scoring [21]. 

Thus, there is growing interest in developing molecular methods to evaluate the HER2/*ERBB2* status. 

In addition to standard solid biopsy, some studies also approached the liquid biopsy strategy in order to overcome the challenge of intra-tumor heterogeneity [23] and to understand the mechanisms of resistance during trastuzumab therapy [28]. Indeed, both these aspects could partially explain the failure of several trials. As shown by Kato et al., *ERBB2* is one of the most frequent altered genes in GEA liquid biopsies and its alterations are significantly associated with poor OS. Moreover, the co-occurrence of alterations in at least one other gene, including *FGFR2*, *RAF1*, *PIK3CA* and *KRAS*, was also observed, and this scenario could be associated with the onset of resistance to anti-HER2 therapy [45]. However, data obtained in both solid and liquid biopsies need further investigation [22,33,34,35]. 

In our study, we used ddPCR to evaluate *ERBB2* amplification in FFPE specimens from both retrospective and prospective cohorts of GEA patients, and cfDNA samples from the prospective cohort. 

In the retrospective cohort, we found that ddPCR can detect *ERBB2* amplification in a larger fraction of GEA (45.3%) than current diagnostic typing based on IHC/SISH (15.1%). As previously reported [33], the HER2 3+ patients showed good concordance. The discordant cases were all HER2 2+ and 0–1+, highlighting the inadequacy of the current diagnostic typing for samples with a low HER2 scoring, and the potential usefulness of the ddPCR method. Furthermore, the adequacy of the cut-off used to identify positive cases was validated by ROC curve analysis.

Similar data were obtained using ddPCR to analyze FFPE-DNAs from the prospective cohort, although the frequency of *ERBB2* amplification in HER2 2+ and 0–1+ cases was lower. This finding is probably due to the relatively smaller number of patients enrolled in the prospective cohort. 

It is interesting to note that when we looked at the time-matched cfDNA samples, we found that a considerable number of cases that were HER2 negative by both IHC/SISH and ddPCR in FFPE-DNA were *ERBB2* amplified with CNV values comparable to those of HER2 positive cases.

Previous studies on GEA patients did not find a higher detection rate of *ERBB2* amplification in liquid biopsy compared to solid biopsy [33,35]. However, Kinugasa et al. [33] performed their study with cfDNA extracted from serum, which could be more contaminated with normal DNA fraction than cfDNA extracted from plasma [46]. A high normal DNA fraction could mask positivity for *ERBB2* amplification in the cfDNA samples. Moreover, both Kinugasa et al. [33] and Kim et al. [35] used different reference genes than ours to set up the cut-off value for *ERBB2* amplification. More intriguing are the data reported by Shoda et al. [22], who set a lower cut-off for *ERBB2* positivity with the same reference gene that we used (*RPPH1*), and found a lower number of positive cases in the cfDNA samples collected just before surgery.

In this study, we found that the HER2 overexpression/*ERBB2* amplification might be underestimated in patients HER2 2+ and 0–1+ using only IHC/SISH and that ddPCR analysis allows for a better stratification of GEA patients, particularly if the analysis is carried out in cfDNA. The high number of positive samples found in liquid biopsy suggests that this approach may better represent the whole tumor genetic landscape than a single solid biopsy and could be very advantageous for tumors with high intra-tumor heterogeneity, such as GEAs. The introduction of cfDNA analysis alongside standard analysis could prevent patients from being incorrectly excluded from targeted therapy, with a potential advantage especially for inoperable locally advanced and metastatic cases.

The benefit of a liquid biopsy approach is emphasized by the data obtained from longitudinal cases. Indeed, in all the patients examined, we found a positivity for *ERBB2* amplification in cfDNA samples collected just before surgery, although some of them were negative in time-matched FFPE specimens. Moreover, we observed that *ERBB2* CNV values increased during the clinical follow-up of patients with disease progression, while a reduction was detected in “non-progressors”. Similar to other studies [47,48], the fluctuation of the standard tumor biomarkers S-CEA and S-CA19.9 did not always reflect the clinical status of patients, highlighting the need to find other more specific biomarkers. 

Our findings on the dynamic monitoring of *ERBB2* amplification in cfDNA are in agreement with previous longitudinal studies that found an increasing CNV at progression in advanced GEA [34,49]. Interestingly, they also found that CNV decreased during trastuzumab treatment and increased again in case of progression. 

Based on our results from serially-collected cfDNAs, it seems that *ERBB2* could be a promising marker to predict the progression of GEAs.

One of the most crucial limitations of this study is that the number of patients in the prospective cohort is relatively restricted. This study lacks also data concerning the trend of *ERBB2* amplification during trastuzumab treatment. Indeed, none of our prospective patients received targeted therapy due to their HER2 negative phenotype, as determined by IHC/SISH, or the presence of comorbidities. The enrollment of other patients is needed to confirm the real impact of the *ERBB2* amplification assessment in cfDNA on clinical management.

Moreover, it would be very useful to design a clinical study based on inoperable locally advanced, and metastatic GEA patients to define the optimal CNV *ERBB2* value cut-off in liquid biopsy for the selection of cases who could truly benefit from trastuzumab targeted therapy.

## 5. Conclusions

Our findings suggest that ddPCR analysis of *ERBB2* amplification in solid biopsy is more sensitive than IHC/SISH in detecting HER2/*ERBB2* overexpression/amplification.

Moreover, we showed that ddPCR, when combined with the more comprehensive approach of liquid biopsy, can increase the likelihood of detecting HER2 positive patients, thus providing them with the possibility to benefit from targeted therapy with trastuzumab.

## Figures and Tables

**Figure 1 cancers-14-02180-f001:**
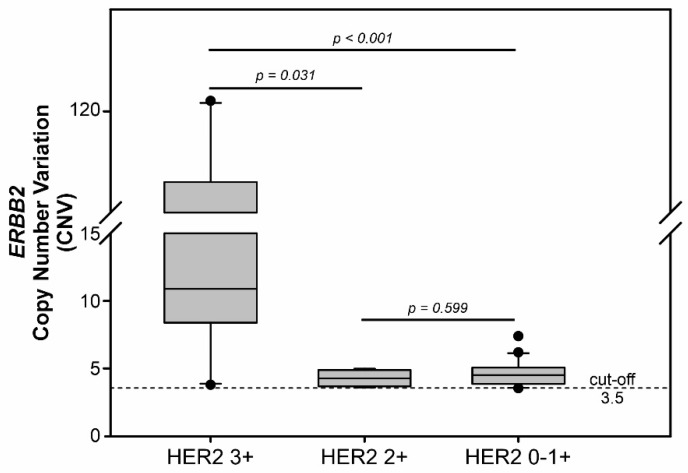
The box plot shows the *ERBB2* CNV positive value distribution in the FFPE-DNA of HER2 3+, HER2 2+, and HER2 0–1+ GEA subgroups (retrospective cohort). Statistically significant differences were present between HER2 3+ and HER2 2+ or HER2 0–1+ subgroups. No difference was found when comparing HER2 2+ and HER2 0–1+ patients. The *p* values were calculated using a two-tailed Wilcoxon Mann-Whitney test; a *p*-value < 0.05 was considered statistically significant. CNV, copy number variation.

**Figure 2 cancers-14-02180-f002:**
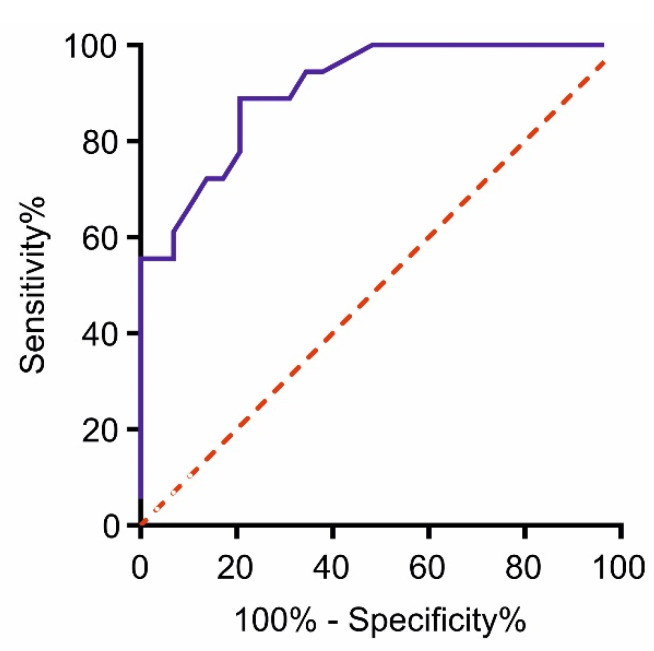
Receiver Operating Characteristics (ROC) curve obtained by comparing *ERBB2* CNV values of the HER2 positive patients vs. CNV values of the HER2 negative ones. HER2 positive or negative status was determined using IHC/SISH staining as “standard method”. Area under the curve (AUC) 0.909, Std. Error 0.04091, CI 95% 0.8288–0.9892, *p*-value < 0.0001.

**Figure 3 cancers-14-02180-f003:**
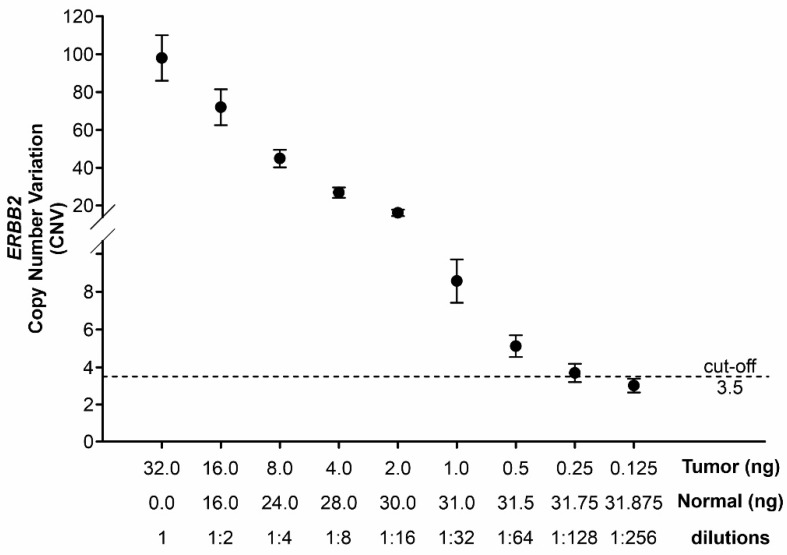
*ERBB2* CNV values in serial dilutions of tumor DNA with increasing amount of its normal counterpart. For each dilution, the absolute quantity (ng) of tumor and normal DNA as well as dilutions were reported. The positivity for *ERBB2* amplification was detected up to a 1:64 dilution.

**Figure 4 cancers-14-02180-f004:**
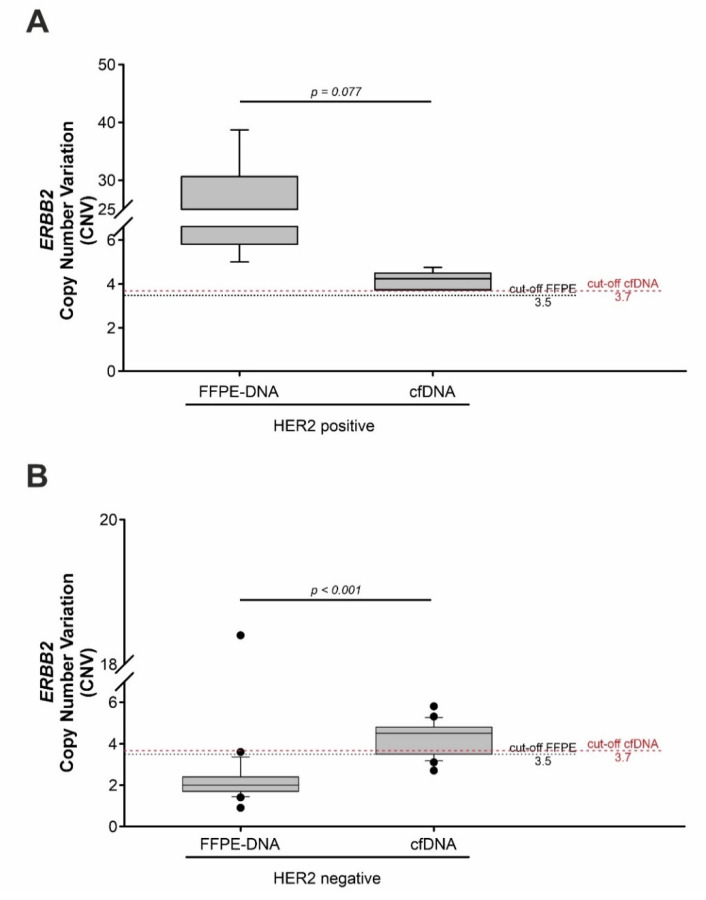
The box plot shows *ERBB2* CNV value distribution in FFPE-DNA and time-matched cfDNA samples in the HER2 positive subgroup (**A**) and in the HER2 negative subgroup (**B**) in the prospective cohort. (**A**) No difference was observed when comparing the FFPE-DNA and cfDNA of HER2 positive patients. The *p*-value was calculated using a two-tailed Student’s *t*-test. (**B**) A statistically significant difference was present between the FFPE-DNA and the cfDNA of the HER2 negative group. The *p*-value was calculated using a two-tailed Wilcoxon Mann-Whitney test. A *p*-value < 0.05 was considered statistically significant. FFPE, formalin-fixed paraffin-embedded; cfDNA, cell-free DNA.

**Figure 5 cancers-14-02180-f005:**
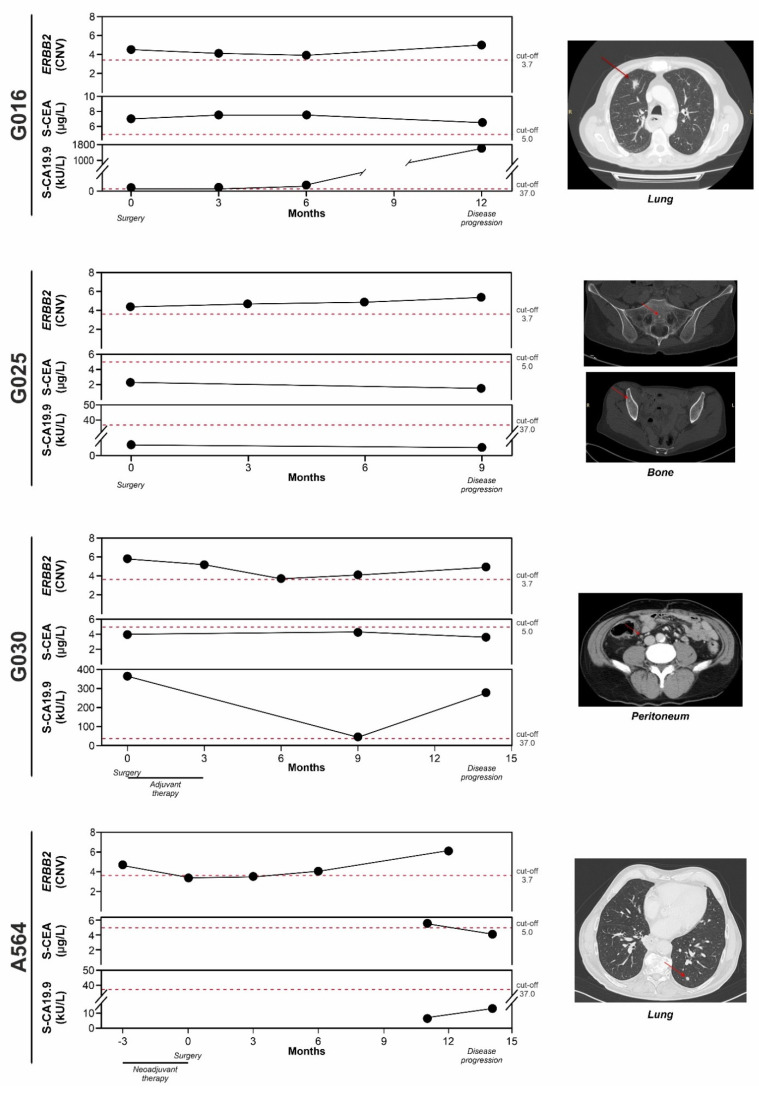
Longitudinal analysis of four “progressor” GEA patients (G016, G025, G030, and A564). *ERBB2* CNV values and tumor markers S-CEA and S-CA19.9 are reported for each patient at different time points. The administration of neoadjuvant or adjuvant therapy, surgery, and the time of disease progression are indicated. A representative TAC image is shown on the right-hand side. A red arrow marks the site of metastasis. S-CEA, soluble carcinoembryonic antigen; S-CA19.9, soluble carbohydrate antigen.

**Figure 6 cancers-14-02180-f006:**
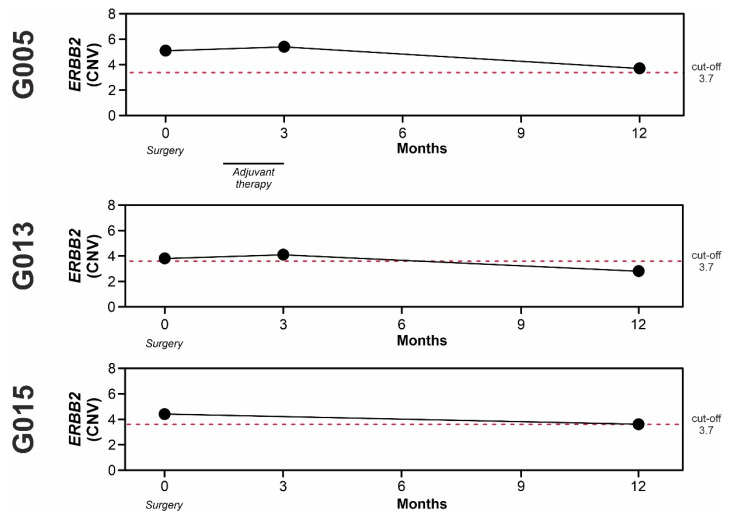
Longitudinal analysis of three “non-progressor” GEA patients (G005, G013, and G015). *ERBB2* CNV values are reported for each patient at different time points. The administration of adjuvant therapy and surgery is indicated.

**Table 1 cancers-14-02180-t001:** Clinicopathologic characteristics of the retrospective cohort of gastroesophageal adenocarcinoma (GEA) patients.

Retrospective Cohort	
**Patients**	*N* (%) 86 (100)
**Age**	
Median (Q1; Q3)	76 (68; 82)
(Range)	(44–97)
**Gender**	
Male	52 (60)
Female	34 (40)
**pTNM stage**	
I/II	48 (56)
III/IV	38 (44)
**IHC Typing**	
EBV+	3 (3)
MSI	15 (17)
CIN	28 (33)
GS	40 (47)
**HER2 status**	
Positive (HER2 3+; HER2 2+/SISH^pos^)	13 (15.1)
Negative (HER2 2+/SISH^neg^; HER2 0–1+)	73 (84.9)

Q1, first quartile; Q3, third quartile; IHC, immunohistochemistry; EBV, Epstein-Barr virus; MSI, microsatellite instability; CIN, chromosomal instability; GS, genomic stable; SISH, silver in situ hybridization.

**Table 2 cancers-14-02180-t002:** HER2/*ERBB2* status of retrospective patients by IHC/SISH and droplet digital PCR (ddPCR).

Retrospective Cohort (86 Patients)			
IHC		SISH	ddPCR
Status/Score	N° Patients	N° *ERBB2* Amplified	N° *ERBB2* Amplified
Positive/3+	11	n.r.	11
Equivocal/2+	7	2	4
Negative/0–1+	68	n.r.	24
Total positive (%)	13/86 (15.1)		39/86 (45.3)

n.r., not required; ddPCR, droplet digital PCR.

**Table 3 cancers-14-02180-t003:** Clinicopathologic characteristics of the prospective cohort of GEA patients.

Prospective Cohort	
**Patients**	*N* (%) 28 (100)
**Age**	
Median (Q1; Q3)	67 (55.3; 74.8)
(Range)	(40–96)
**Gender**	
Male	17 (60.7)
Female	11 (39.3)
**pTNM stage**	
I/II	9 (32.1)
III/IV	19 (67.9)
**IHC Typing**	
EBV+	0 (0)
MSI	2 (7.1)
CIN	8 (28.6)
GS	18 (64.3)
**HER2 status**	
Positive (HER2 3+; HER2 2+/SISH^pos^)	5 (17.9)
Negative (HER2 2+/SISH^neg^; HER2 0–1+)	23 (82.1)

**Table 4 cancers-14-02180-t004:** HER2/*ERBB2* status in FFPE-DNA and cfDNA of prospective patients based on IHC/SISH and ddPCR.

Prospective Cohort (28 Patients)				
IHC		SISH	ddPCR	
			FFPE-DNA	cfDNA
Status/Score	N° Patients	N° *ERBB2* Amplified	N° *ERBB2* Amplified	
Positive/3+	5	n.r.	5	5
Equivocal/2+	7	0	1	5
Negative/0–1+	16	n.r.	1	11
Total positive (%)	5/28 (17.9)		7/28 (25%)	21/28 (75%)

FFPE, formalin-fixed paraffin-embedded; cfDNA, cell-free DNA.

## Data Availability

The data presented in this study are available from the corresponding author on request.

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
