# Peer review of "Putative Clinical Potential of ERBB2 Amplification Assessment by ddPCR in FFPE-DNA and cfDNA of Gastroesophageal Adenocarcinoma Patients"

_cancers, 2022, doi:10.3390/cancers14092180_

Round 1

Reviewer 1 Report

After revision, the authors improved their manuscript.
Overall, it is difficult to follow the authors’ revision, as there are no pages and lines mentioned, where the appropriated changes were made.
Comments:

Reply of the authors: HER2 staining and FISH are now standardized methods and we believe that they will not add relevant information. However, below we show to the Reviewer a Figure reporting the staining.

The revision is not only to satisfy the reviewer. As seen in the VARIANZ trial, “HER2 staining and FISH” might not be standardized methods as they should. Therefore, please include a representative figure for Her2-neg, 1/2/3 at least to the supplemental data.

Reply of the authors: The quality assessment of cfDNA using Agilent Tape Station 2200 is a routinely used approach. However, below we show to the Reviewer an image of our cfDNAs showing that the majority of our samples are of good quality, not contaminated with germline DNA. In the few cases of a contamination (G13 and G20), the samples resulted positive for ERBB2 amplification. Germline contaminated cfDNA should be discarded only in case of negative result.

Again, the comments from the reviewer are to improve the manuscript. Please, include the results to the supplemental data. I do not agree to the author’s opinion that quality assurance should not be shown in scientific reports. It is very important that all results, which were obtained are shown. This will increase transparency and readers are able to judge about the presented data more precisely.
“showing that the majority of our samples are of good quality” must be mentioned in the results together with the raw data (maybe as supplemental data).
I am not able to find the additional results in regard to ROC analysis, which are mentioned by the authors in the response letter.
“However, we draw a ROC curve for the retrospective FFPE DNA samples and we compare the ERBB2 CNV results (ddPCR) with the “standard methods” based on IHC/SISH. Patients were categorized as positive (HER2 3+ in IHC or HER2 0-1+-2+ in IHC and amplified in SISH) or negative (HER2 0-1+-2+ in IHC and not amplified in SISH). We obtained a ROC curve area of 0,909±0,041 and a p-value statistically significant (< 0.0001). By assigning a cut-off value ≥ 3.5 (the same that we calculated as described in material and methods), ddPCR analysis showed a sensitivity of 100 % and a specificity of 51.72 %. These results make us confident on the reliability of the cut-off defined for cfDNA analysis (≥ 3.7).”

My following concerns are not answer appropriated:

I am also not convinced, that the CNV variation in the longitudinal studies can predict anything. Why the authors only isolated cfDNA from four patients in the longitudinal study. All four patients had a disease progression. To confirm the relevance, the authors need to show the decrease of CNV in cured
patients to levels of healthy controls.

The answer of the authors is minimalistic and it is not clear to me, why the authors only investigated progressing patients? Including a further repletion of the own results to the discussion is not appropriated:

Discussion

“This point is emphasized by data from longitudinal cases. Indeed, all these patients showed positivity for ERBB2 amplification in their cfDNA samples collected just before surgery. Moreover, we observed that ERBB2 CNV values increased during the clinical follow-up of patients with disease progression,
while a reduction was detected in “non-progressors”. Based on the results from serially-collected cfDNAs, it seems that ERBB2 could also be a promising marker to predict the progression of GEAs.”

The discussion must be still improved, there is only one new ref. (yellow) in the revised version of the manuscript. I really encourage the authors, to have a deep look to the latest literature in regard to Her2 in GC and EAC. This is mandatory, before the manuscript can be considered for publication.

Many paragraphs of the discussion are still a repletion of the authors’ results.

Reviewer 2 Report

This manuscript has been improved by responding reviewer’s many questions. Additive detailed information of HER2/ERBB2 status and clinicopathologic analysis in the retrospective cohort is useful for readers. I think this manuscript is worth for publication in Cancers.

Author Response

Thanks for the acceptance of our manuscript.